# A Homogeneous Colorimetric Strategy Based on Rose-like CuS@Prussian Blue/Pt for Detection of Dopamine

**DOI:** 10.3390/s23229029

**Published:** 2023-11-07

**Authors:** Di Yang, Jiao Ran, Huafei Yi, Pujin Feng, Bingqian Liu

**Affiliations:** Guizhou Engineering Laboratory for Synthetic Drugs (Ministry of Education of Guizhou Province), College of Pharmacy, Guizhou University, Guiyang 550025, China

**Keywords:** colorimetric, dopamine, rose-like CuS@Prussian blue/Pt, smartphone

## Abstract

The development of effective methods for dopamine detection is critical. In this study, a homogeneous colorimetric strategy for the detection of dopamine based on a copper sulfide and Prussian blue/platinum (CuS@PB/Pt) composite was developed. A rose-like CuS@PB/Pt composite was synthesized for the first time, and it was discovered that when hydrogen peroxide was present, the 3,3′,5,5′-tetramethylbenzidine (TMB) changed from colorless into blue-oxidized TMB. The CuS@PB/Pt composite was characterized with a scanning electron microscope (SEM), an energy dispersive spectrometer (EDS), and an X-ray photoelectron spectrometer (XPS). Moreover, the catalytic activity of the CuS@PB/Pt composite was inhibited by the binding of dopamine to the composite. The color change of TMB can be evaluated by the UV spectrum and a portable smartphone detection device. The developed colorimetric sensor can be used to quantitatively analyze dopamine between 1 and 60 µM with a detection limit of 0.28 μM. Furthermore, the sensor showed good long-term stability and good performance in human serum samples. Compared with other reported methods, this strategy can be performed rapidly (16 min) and has the advantage of smartphone visual detection. The portable smartphone detection device is portable and user-friendly, providing convenient colorimetric analysis for serum. This colorimetric strategy also has considerable potential for the development of in vitro diagnosis methods in combination with other test strips.

## 1. Introduction

Dopamine (DA) is an important neurotransmitter that plays a key role in neuropsychiatric illness [1]. DA is an important and abundant monoamine neurotransmitter that is closely connected to many neuronal activities, such as cognitive function [2], motor control [3], and memory function [4]. In particular, the pathophysiology of several psychiatric and neurological conditions, including Parkinson’s disease [3], Huntington’s disease [5], food and drug addiction [6], and schizophrenia [7], is intimately linked to aberrant DA secretion. Parkinson’s disease is the second most common neurodegenerative disease; however, the early pathophysiological events and sequences of its dysfunction remain unknown. The loss of dopaminergic neurons and lower levels of striatal DA can lead to the motor impairments in Parkinson’s disease [3]. Therefore, the development of an efficient, consistent, and sensitive method of detecting DA is crucial.

Recently, several techniques have been developed for DA detection, including high-performance liquid chromatography (HPLC) [8], liquid chromatography-mass spectrometry (LC-MS) [9], electrochemical methods [10,11,12], fluorescence spectrophotometry [13], and colorimetric sensors [14]. These methods have demonstrated advantages in terms of accuracy and precision. However, these methods, more or less, have some weaknesses. For example, the HPLC and LC-MS have disadvantages such as time consumption, expensive instruments, and operator training. Electrochemical detection has been widely noticed because of its low cost, fast detection, high sensitivity, and simple operation, but it is relatively less intuitive. The colorimetric assay has the advantages of simplicity, rapidity, and intuitiveness. It has also been reported that an electrochemical-colorimetric dual-mode detection of dopamine has been established [14]. Moreover, there have also been reports of colorimetric detection using smartphones by capturing the color changes without the need for additional tools. Colorimetric sensors are based on enzymes that change their color and absorbance during DA detection. Since these sensors are inexpensive, portable, rapid-acting, and clearly visible to the naked eye, they have been extensively utilized for DA detection [15,16,17].

Nowadays, many nanomaterials are known to exhibit enzyme-like activity. These nanomaterials include precious metal nanoparticles [18], metal oxides [19], metal sulfides, carbon nanotubes [20], graphene and derivatives [21], metal-organic frameworks [22], etc. Recently, precious metal nanomaterials such as platinum nanoparticles (NPs) have been considered promising candidates for application in colorimetric sensors owing to their peroxidase activity [23]. However, they regularly group together in catalytic reactions, which decreases the catalytic activity. Currently, an efficient solution to this issue is the hybrid nanomaterials created by growing Pt NPs on substrates such as graphene oxide, MoS_2_ nanosheets, MoO_3_ nanosheets, and Pd nanosheets [24,25,26,27]. More importantly, these hybrid nanomaterials’ catalytic activity can be significantly increased due to the synergistic interaction between their bimetallic active centers. Cu-based nanomaterials such as copper oxide (CuO) [28]and copper phosphate Cu_3_(PO_4_)_2_ [29] have received a great deal of attention for sensing and catalysis-related applications [30]. Due to their enzymatic-like behavior for the catalyzed oxidation of per-oxidase substrates in the presence of H_2_O_2_, copper sulfide (CuS)-based nanomaterials have been shown to be promising nanozymes for glucose, cholesterol, and uric acid sensing [31]. A CuS-based sensor for sensing dopamine has previously been reported in conjunction with rGO [32]. Among them, Prussian blue (PB) has attracted a lot of study interest because of its significant peroxidase mimic catalytic activity, which is related to its mixed valence states of Fe atoms, which are similar to Fe_3_O_4_ [33,34]. Due to their excellent peroxidase mimic characteristics, they are commonly employed as transducers in the development of electrochemical biosensors [34]. Prussian blue (PB) NPs with high enzyme-mimicking activity supported by MoS_2_ nanocomposites have been reported as peroxidase-like nanozymes for the colorimetric sensing of DA [35,36,37]. Compared with single nanomaterials, multi-component systems may be able to solve the above problems and can generally exhibit high enzymatic simulation catalytic activity.

In this study, we developed a rapid, homogeneous colorimetric method based on copper sulfide Prussian blue/platinum (CuS@PB/Pt) NP composites for DA detection. The colorimetric performance of the NPs was improved by the addition of Pt. To effectively replace natural enzymes like horseradish peroxidase (HRP) in the chromogenic reaction between H_2_O_2_ and 3,3′,5,5′-tetramethylbenzidine (TMB), the CuS@PB/Pt NP composites were synthesized. DA successfully prevented the chromogenic reaction by oxidizing colorless TMB to blue oxidized TMB (ox-TMB), effectively turning off the colorimetric signals. This assay was validated for the detection of DA in human serum. This method provides a strategy for DA identification and may be suitable for practical application.

## 2. Materials and Methods

### 2.1. Reagents and Apparatus

The details are described in the Appendix A.

### 2.2. Synthesis of CuS@PB/Pt

The CuS NPs and CuS@PB composites were prepared using methods described by Qu et al. [38] and Li et al. [39], whereas the CuS@PB/Pt composite was prepared using a method described by Gong et al. [28]. The Appendix A contain a detailed description of the preparation procedures.

### 2.3. Colorimetric Detection of DA

In PBS buffer (0.01 M, pH = 7, 320 μL), different concentrations of DA aqueous solution were mixed with CuS@PB/Pt (1 mg/mL, 30 μL). Then, TMB (50 mM dissolved in ultrapure water, 110 μL) and H_2_O_2_ (50 mM, 50 μL) were added to the above solution. After 16 min of incubation at 45 °C, the mixture was subjected to UV-vis absorption spectroscopy to measure the absorbance. A calibration curve was prepared by plotting the absorbance at 652 nm as a function of the DA concentration (A = A_0_ − A, where A_0_ and A are the absorbance values at 652 nm in the absence and presence of DA, respectively). We took pictures immediately, and the RGB value of the color change of the reaction system was extracted using the built-in chromaticity extraction software (Color extractor, Android app market). The *R* value was used to quantitatively determine the DA concentration in the system. A portable smartphone colorimetric detection device was developed according to the method reported by Qu et al. [40]. We made a slight improvement by adding a cover to the device to avoid affecting the stability of the method due to a change in the light source. A schematic of the device is shown in Figure 1c.

### 2.4. Quantitative Analysis of DA in Human Serum Samples

To successfully detect DA in human serum, samples from healthy volunteers were obtained from the Guizhou Staff Hospital (Guizhou, China). Before use and after thawing at room temperature, the samples were centrifuged (6708× *g*) for 10 min to precipitate macromolecular proteins and other impurities, and the supernatant was diluted 50× for testing. In addition, different concentrations of DA standard solution (2–30 ng/mL) were added to the prepared serum samples and measured using UV-vis absorption spectroscopy and a smartphone. The recovery rate was calculated using the following formula:(1)$$Recovery\left(\%\right)=\frac{C_{detected}-C_{added}}{C_{added}}\times 100\%$$

## 3. Results

### 3.1. Characterization of Materials

To ensure the success of the experiment, it was important to determine whether the CuS@PB/Pt composite had been properly synthesized. First, the morphologies of CuS, CuS@PB, and CuS@PB/Pt were characterized using SEM. As shown in Figure 2a, the CuS NPs exhibited the shape of a rose composed of nanosheets. The thickness of the nanosheets was uniform and clear [38]. The CuS@PB (Figure 2b) changes in morphology mean that the thickness of the nanosheets was thicker. As shown in Figure 2c, the CuS@PB/Pt composite recovered its flower shape, and small particles were present on the nanosheets. In addition, the EDS results revealed the presence of S, Fe, Cu, and Pt in the CuS@PB/Pt composite (Figure 2d,h,l).

X-ray photoelectron spectroscopy (XPS) was used to further investigate the chemical composition of the CuS@PB/Pt composites. As shown in Figure 2g, there were two main characteristic peaks at 930.97 and 950.78 eV in the Cu2p spectrum, which were assigned to Cu 2p3/2 and Cu 2p1/2, respectively. Peaks corresponding to S 2p appeared at 160.88 eV (S 2p3/2) and 162.13 eV (S 2p1/2) [39] (Figure 2i). The Pt 4f profile showed two peaks at 71.63 and 76.53 eV, which correspond to metallic Pt 4f7/2 and Pt 4f5/2, respectively. Pt mainly exists in the zero-valence and 2^+^-valence forms. Three peaks in the Fe 2p spectrum (Figure 2k), which correspond to Fe 2p3/2, are evident at approximately 708.6, 709.6, and 711.53 eV [39].

### 3.2. Feasibility of the Designed Colorimetric Strategy

Evaluation of the oxidase activity of the CuS@PB/Pt composite is essential because it directly influences the TMB signal response. First, we studied the oxidase activity of the synthesized materials using UV-vis absorption spectroscopy. TMB, the most widely used chromogenic substrate in nanoenzyme catalysis, was used as a chromogenic reagent in this experiment. Typically, CuS@Pb/Pt (1 mg/mL, 30 μL) was dispersed into PBS buffer (0.01 M, pH = 7, 320 μL) and ultrapure water (480 μL). TMB (50 mM, 110 μL) and H_2_O_2_ (50 mM, 50 μL) were added to the solution. The TMB was dissolved in ultrapure water. The mixture was incubated at 37 °C for 10 min. After that, the absorbance data were immediately collected using UV-vis absorption spectroscopy. The peroxidase-like catalytic activities of CuS, CuS@PB, and CuS@PB/Pt (1.0 mg/mL) were investigated. As shown in Figure 3a, the CuS@PB/Pt composites exhibited a high absorbance of 652 nm compared to the other components, such as CuS and CuS@PB. Moreover, except when H_2_O_2_ and the CuS@PB/Pt composite existed at the same time, the absence of an absorption peak at 652 nm in the other components further demonstrates the superiority of the composite over the alternatives (Figure 3b). This phenomenon agrees with previous reports that H_2_O_2_ would first be quickly catalyzed into hydroxyl radicals in the presence of peroxidase, which would then further oxidize the colorless TMB to a blue ox-TMB [41].

In addition, we investigated the oxidase activity of the CuS@PB/Pt composite using a color-producing process. The CuS@PB/Pt composite oxidized colorless TMB to blue ox-TMB in the presence of hydrogen peroxide. The oxidative activity was demonstrated by Michaelis–Menten kinetics (*v* = *V_max_*[S]/(*K_m_* + [S]), where [S] is the concentration of the substrate, *v* is the initial velocity, *K_m_* is the Michaelis–Menten constant, and *V_max_* is the maximal reaction velocity.

As shown in Appendix A, CuS@PB/Pt had a *K_m_* value of 3.5 mM when H_2_O_2_ was used as the substrate, which was less than that previously observed for HRP (3.70 mM) [42]. Unlike HRP, which has a *K_m_* value of 0.434 mM, the CuS@PB/Pt nanocomposites have a *K_m_* value of 0.298 mM (Appendix A) with TMB as the substrate. In addition, the feasibility of the proposed colorimetric strategy for the detection of DA was explored. As shown in Figure 3c, when DA was added in various quantities, the ox-TMB absorption peak at 652 nm decreased in intensity (10 μM, 100 mΜ). To further investigate the feasibility of the suggested colorimetric strategy, the oxidative reactions of other chromogenic reagents such as 2,2′-azino-bis(3-ethylbenzothiazoline-6-sulfonic acid) (ABTS) and *o*-phenylenediamine (OPD) were also performed (Appendix A). The characteristic absorbance of the UV-vis spectra (Appendix A) indicates that the colorimetric strategy for DA detection was successfully developed.

### 3.3. Optimization of Experimental Parameters

HRP and other peroxidase mimics are comparable to natural peroxidases, and the pH and temperature of the experiment can significantly impact the catalytic ability of the CuS@PB/Pt composites. To investigate how pH affects the catalytic oxidation of TMB by CuS@PB/Pt in the presence of H_2_O_2_, we conducted pH experiments using buffer solutions with different pH values. In Figure 4a, the changes in the peroxidase-like activity of the CuS@PB/Pt composites are shown at different pH levels (3.0–8.0). The findings show that the best catalytic activity was achieved at approximately pH 7. Therefore, a pH of 7 was selected for the ensuing test. Additionally, a broad temperature range (20–60 °C) was examined to find out how temperature affected the catalytic activity of CuS@PB/Pt. The catalytic activity of CuS@PB/Pt was found to be at its optimum level at 45 °C (see Figure 4b). These results are consistent with those of Figure 3d, which suggests that the relative activity of CuS@PB/Pt is maintained above 40% over the temperature range of 20 °C to 60 °C. The response time of the CuS@PB/Pt composite to ox-TMB was also investigated. When the CuS@PB/Pt composites interacted with TMB, the UV-vis spectra of ox-TMB tended to be maximal and constant after 16 min of reaction, as shown in Figure 4c.

### 3.4. Analytical Performance of Colorimetric Detection of DA

To evaluate the sensing capacity of CuS@PB/Pt for the colorimetric detection of DA, various concentrations (1–60 µM) of DA were measured under the optimal conditions. As shown in Figure 5a, the absorbance decreased progressively as the DA content increased, which is compatible with the color change shown in Figure 5c. There were two linear ranges at low (1–10 µM) and high concentrations (10–60 µM) of DA. The following regression equation applies: y_1_ = −0.011x_1_ + 0.8934 (R^2^ = 0.9965), where x_1_ is the concentration of 1–10 µM; y_2_ = −0.031x_2_ + 1.119 (R^2^ = 0.993), where x_1_ is the concentration of 10–60 µM, and the detection limit is 0.28 μM. In addition, we captured the color changes on a smartphone. According to the *R* value produced by the smartphone, the DA concentration has a linear relationship with color at low concentrations (1–10 µM). The linear regression equation is as follows: y = 6.5182x–0.6515 (R^2^ = 0.9921), and 0.42 μM is the detection limit. The linear concentration range and detection limit for DA measurement in this study are consistent with those of previous DA detection methods (Table 1). These results have the advantages of a short detection time, the use of portable equipment, and detection by smartphone.

### 3.5. Selectivity, Reproducibility, and Stability of the Colorimetric Method

Several common ions in human serum were studied under the same conditions to assess the selectivity of the novel colorimetric technique for DA detection. As shown in Figure 6a, only DA can drastically reduce the signal intensity. However, although their concentration was five times greater than that of DA (250 µM), the other ions seldom led to a discernible loss in signal intensity. As shown in Figure 6d, the same result was obtained using the smartphone to measure the *R* value. At the same time, Figure 6a,e shows that the absorbance of the other interferents, with the exception of AA, remains almost unchanged in the colorimetric sensor. Therefore, we introduced a masking agent, iodoacetamide (IAM), to address this issue. The introduction of IAM shields the interfering signal response well, thereby improving the selectivity of the sensor for DA, as shown in Figure 6b,f. Additionally, we evaluated the reproducibility of the suggested method using intra- and inter-batch experiments. Three different concentrations of DA (5 µM, 10 µM, and 30 µM) were tested using the same batch of synthetic CuS@PB/Pt composite. The relative standard deviation (RSD) corresponding to the absorbance values of three parallel experimental groups was less than 0.80%. The RSD values corresponding to *R* with three different concentrations (2 µM, 5 µM, and 10 µM) were less than 0.01%. Three batches of CuS@PB/Pt composites were used to detect DA samples (10 µM). The RSD values were below 1.19% and 0.8%, respectively. Additionally, under the same experimental conditions, samples prepared at the same DA concentration were tested seven times to evaluate the repeatability of this method. The UV-vis spectral responses and *R* values of the seven replicates were similar, indicating that the constructed sensor had satisfactory reproducibility (Figure 6b,e). This indicates that the colorimetric method has satisfactory reproducibility. We further examined the stability of the CuS@PB/Pt composite because the system had a significant impact on sample detection. When the response of the CuS@PB/Pt composite was examined after storage at 4 °C for 30 d, the absorbance of 5 µm DA changed rarely. As shown in Figure 6f, the same result was obtained using the smartphone to determine *R*. This demonstrates the excellent stability of the fabricated system. The result indicates that the CuS@PB/Pt composite has the potential for use in long-term applications. to be used for long-term applications.

### 3.6. Real-Sample Analysis

The feasibility and applicability of the sensor for analyzing actual samples were further verified by testing DA added to the human serum samples of healthy volunteers and calculating the recovery rate. As shown in Appendix A, the recovery rate of the standard addition by the UV-vis absorption spectrum was 94.18–105.55%, and the average recovery of the smartphone signal was 88.71–107.58%. The detection results of the two methods were compared with those of HPLC-MS/MS. After mutual fitting, the slope of the linear equation obtained by the two modes was close to 1, and the intercept was close to 0, which indicates that the two detection methods are consistent with HPLC-MS/MS and further proves that the sensor is feasible and accurate for detecting DA (Figure 7).

## 4. Conclusions

In this study, a rose-like CuS@PB/Pt composite was successfully fabricated. Moreover, the excellent peroxidase-like catalytic activity of the composite was observed for the first time. However, the system has economic challenges due to the use of precious metal nanomaterials. The narrow range of detection using smartphones is also a limitation. In spite of this, we developed a homogeneous colorimetric strategy for the measurement of colorimetric signals using UV-vis absorption spectroscopy that can instantly produce smartphone readouts for DA detection. The developed colorimetric sensor can be used to quantitatively analyze dopamine between 1 and 60 µM with a detection limit of 0.28 μM. In addition, we developed an easy-to-use portable device to aid in colorimetric detection by smartphone. The quantitative point-of-care analysis of DA is possible because of the rapid (16 min) detection time and portable smartphone readouts for detection signals. The applicability of the proposed approach for the analysis of actual sample data was demonstrated through the acceptable detection of DA in real samples by the proposed colorimetric sensor. In summary, a practical and effective colorimetric detection platform was developed for the detection of DA in human serum that will broaden the potential point-of-care applications in biotechnology and clinical diagnosis. This platform also provides a good future research direction for detecting dopamine precursors or metabolites, such as phenylalanine and tyrosine. Moreover, the potential application of novel, high-performance artificial enzyme simulations based on nanomaterials was demonstrated.

## Figures and Tables

**Figure 1 sensors-23-09029-f001:**
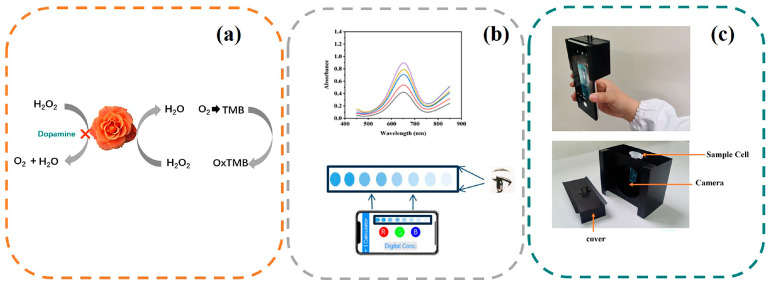
(**a**) Schematic illustration of a CuS@Pb/Pt colorimetric sensor for DA. (**b**) Two methods for detecting dopamine. (Coloured lines for UV-vis absorption spectroscopy; Shades of blue for visible color of DA) (**c**) The schematic representation of the portable device.

**Figure 2 sensors-23-09029-f002:**
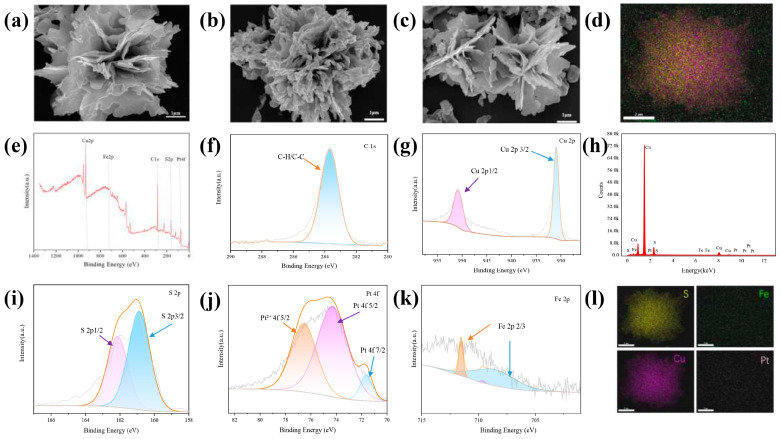
SEM images of (**a**) CuS, (**b**) CuS@PB, and (**c**) CuS@PB/Pt. (**e**) XPS spectra of the CuS@PB/Pt. XPS spectra of (**f**) C 1_S_, (**g**) Cu 2p, (**i**) S 2p, (**j**) Pt 4f, and (**k**) Fe 2p. (**d**,**h**,**l**) The corresponding EDS elemental mapping images of the prepared CuS@PB/Pt, including the S, Cu, Fe, and Pt. Scale bar: 2 nm.

**Figure 3 sensors-23-09029-f003:**
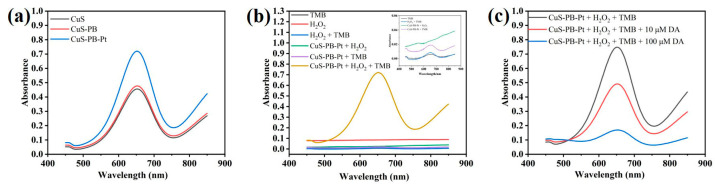
(**a**) Responses of UV-vis spectra of various materials to TMB and H_2_O_2_ at the same concentration (1.0 mg/mL). (**b**) Responses of the UV-vis spectra of CuS@Pb/Pt (1.0 mg/mL) reacted with TMB with or without H_2_O_2_. (**c**) Responses of UV-vis spectra to the addition of DA at different concentrations.

**Figure 4 sensors-23-09029-f004:**
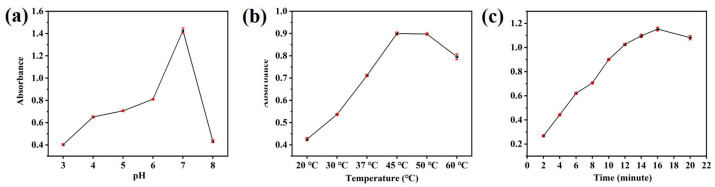
(**a**) UV-vis absorption spectra with different pHs of PBS (0.01 M) buffer. (**b**) UV-vis absorption spectra with different reaction temperatures. (**c**) UV-vis absorption spectra with different reaction times.

**Figure 5 sensors-23-09029-f005:**
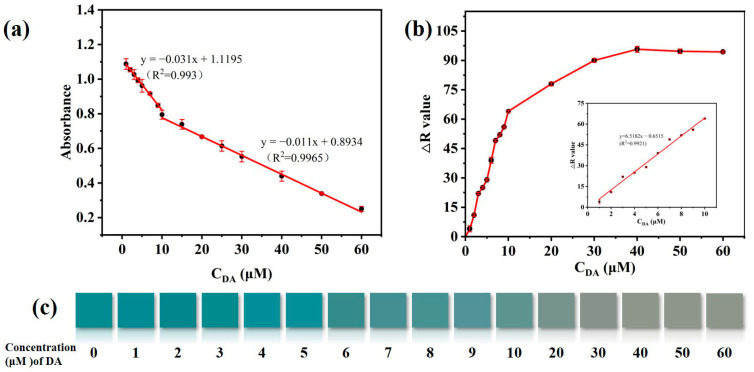
(**a**) Linear curve of UV-vis spectra responses in the DA from 1 to 60 μM. (**b**) Linear curve of RGB colorimetry in the DA from 1 to 10 μM. (**c**) Matching visible color in the DA range of 1 to 60 μM.

**Figure 6 sensors-23-09029-f006:**
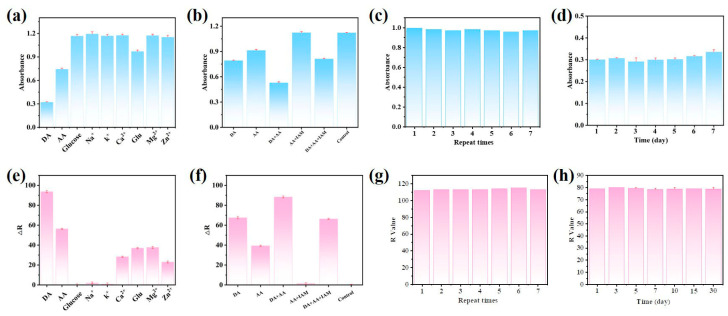
(**a**) Specificity of the established method for the DA of UV-vis spectra responses (other interfering compounds are present in concentrations five times (50 μM) greater than DA’s. (**e**) Specificity of the established method for DA of RGB colorimetric (△R = R − R_blank_). (**b**,**f**) The UV-vis spectra responses and RGB colorimetric values of DA, AA, and coexisting systems. The concentration of interferents is three times higher than that of DA (10 μM). (**c**,**g**) The repeatability of the developed method (DA = 10 µM). (**d**,**h**) The 30-day storage stability of the established method (DA = 50 µM for UV-vis spectra, DA = 5 µM for RGB colorimetric).

**Figure 7 sensors-23-09029-f007:**
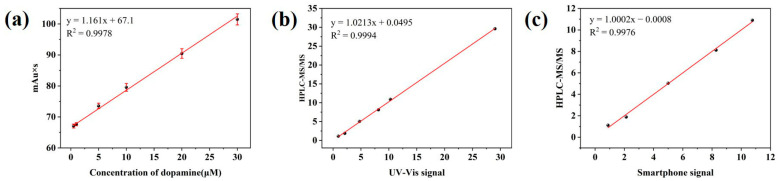
(**a**) Linear curve of peak area versus dopamine concentration. (**b**) Fitting curve of HPLC-MS/MS and UV-vis signal. (**c**) Fitting curve of HPLC-MS/MS and smart phone signal.

**Table 1 sensors-23-09029-t001:** Comparison of different methods for the detection of dopamine.

Method	Materials	LOD	Sensitivity	Ref
HPLC-MS/MS	/	1.87 μM	1.87 × 10^−5^–1.87 μM	[9]
Electrochemical	Co_3_O_4_–Fe_2_O_3_	0.24 µM	10–100 µM	[41]
Fluorescence	2, 3-diaminophenazine	1.76 μM	2.0–61 μM	[13]
Colorimetric	MVCM	0.74 μM	5–100 μM	[42]
Colorimetric	CuS@PB/Pt	0.28 μM	1–60 µM	This work

## Data Availability

The data presented in this study are available on request from the corresponding author.

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
