# Peer review of "A Homogeneous Colorimetric Strategy Based on Rose-like CuS@Prussian Blue/Pt for Detection of Dopamine"

_sensors, 2023, doi:10.3390/s23229029_

Round 1
Reviewer 1 Report
Comments and Suggestions for Authors
I read carefully the research article entitled A simple homogeneous colorimetric strategy based on rose-like CuS@Prussian blue/Pt for detection of dopamine to Sensors. The concept of the article is interesting however the results are very confusing and their usefulness in the development of lignocellulosic biorefineries. This manuscript is generally well written and clearly presented however still needs to address some comments, and thus require moderate revision to improve the quality of the manuscript.
1. Title should modify which can describe the whole research work how authors can say it is simple?
2. Abstract looks very general. Need to rewrite in precise way and at the end mention the importance of research work briefly.
3. In the introduction discuss briefly about recent advances for the detection of dopamine with their pros and cons.
4. Optimization studies results should be in the main text.
5. Is there any effect of temperature and pH on the results?
6. Is there any morphological changes that can be observed with the developed material?
Statistical analysis of the results should be provided in the materials and methods section. It's important for all experimental work Report these values in the results and discussion
7. Compare your results with other studies by adding one table for the same.
8. Techno Economic challenges and limitations of this system should be included? Add future research directions also.
9. The conclusion of the study needs to be added with the specific output obtained from the study, it could be modified with precise outcomes with a take home message.
10. Some English and grammar mistakes are present that need to be correct to improve the quality of the manuscript.
Comments on the Quality of English LanguageSome English and grammar mistakes are present that need to be correct to improve the quality of the manuscript.
Author Response
Comments:
1.The introduction in my opinion should be expanded. There should be more discussion of the presence of other dopamine-detecting sensors in the literature for more comparison with your sensor (e.g. 10.3390/chemosensors11070379, 10.1149/1945-7111/acb237,10.1016/j.electacta.2022.141535, etc.). I would also extend the discussion to get a full picture of the importance of detecting this neurotransmitter. For example, I would also discuss the interest of researchers in trying to detect dopamine precursors or metabolites such as phenylalanine (10.1109/TIM.2023.3284027, 10.1002/elan.202200501, etc.), and tyrosine (10.1016/j.mtcomm.2023.106036,/10.1016/j.bios.2023.115360,etc)
Re: Thank you for your suggestion. We have expanded on other dopamine-detecting sensors in introduction, as shown in the text marked in red. I would also extend the discussion to examine dopamine precursors or metabolites such as phenylalanine and tyrosine as a future direction of research.
2.Re-check the text for some writing errors.
Re: We sincerely thank the reviewer for careful reading. In our resubmitted manuscript, the typo is revised. Thank you for your correction.
3.Line 68 " The Supporting Information contains a detailed 68 description of the preparation procedures." Unfortunately, I do not see the supporting information loaded. Entered in file.
Re: We are very sorry that you did not see our supporting literature. We have compressed the manuscript and supporting documents and uploaded them to the website at the time of submission. You can see the method of material synthesis in the supporting document (“Synthesis of CUS, CuS@PB, CuS@PB/pt”).
4.The description part of the information on the materials used and the instrumentation used is missing. This part must be inserted.
Re: Thanks for your reminding! The information of materials and instrumentation have already showed in supporting materials and we have added the "2.1Reagents and apparatus" section in the manuscript according to your request.
5.Uniform all images of the paper in the graphics. They all have different graphics. For example in figure 2 they look different sizes, the letters inside also look different, the boxes look like some are marked and some are not, etc. Also increase the quality (the lettering inside the XPS cannot be read), Figure 3 the letters here are on the outside of the boxes, in figure 2 they were on the inside, it would be good to standardise this throughout the text, etc.......
Re: Thanks for your careful checks. We are sorry for our carelessness. Based on your comments, we have unified the size of the picture and the position of the letters. In addition, we have improved the quality of XPS images in figure 2.
7.Section 3.2 please refer to the support which unfortunately is not there and therefore I cannot evaluate.
Re: We are very sorry that you did not see our supporting literature. We have compressed the manuscript and supporting documents and uploaded them to the website at the time of submission. You can see this part in the supporting document (Figure S1, S2).
Figure S 1. (a)(c)Steady‐state kinetic analysis utilizing the Michaelis–Menten model of H2O2 and TMB; (b)(d) Lineweaver–Burk double‐reciprocal model of H2O2 and TMB.
Figure S2. UV–Vis absorption spectra with different chromogenicreagent of the addition of DA at different concentrations((a)ABTS;(b) (OPD))
8.Also section 3.3 I cannot evaluate it I do not see the supporting.
Re: As described in the previous question, you can see this part in the supporting document (Figure S3).
Figure S3 (a) UV-Vis absorption spectra with different pH of PBS (0.01M) buffer; (b) UV–Vis absorption spectra with different reaction temperature; UV-Vis absorption spectra with different reaction time.
9) I would insert a table comparing your sensor with similar ones in the literature.
Re: Thank you for your suggestion. We have added a table (Table1) for comparison of the analytical parameters of our sensor and previous sensors reported in the literature for the detection of DA, as shown in the text marked in red.
Table 1 Comparison of different methods for detection of dopamine.
|
Method |
Materials |
LOD |
Sensitivity |
Ref |
|
HPLC-MS/MS |
/ |
1.87μM |
1.87× 10−5-1.87μM |
[9] |
|
Electrochemical |
Co3O4–Fe2O3 |
0.24 µM. |
10-100 µM |
[29] |
|
Fluorescence |
2, 3-diaminophenazine |
1.76 μM |
2.0 - 61 μM |
[13] |
|
Colorimetric |
MVCM |
0.74μM |
5-100 μM |
[17] |
|
Colorimetric |
CuS@PB/Pt |
0.28μM |
1 - 60 µM |
This work |

Reviewer 2 Report
Comments and Suggestions for Authors
see attached file

Minor editing of English language required
Author Response
I read carefully the research article entitled A simple homogeneous colorimetric strategy based on rose-like Cus@Prussian blue/Pt for detection of dopamine to Sensors. The concept of the article is interesting however the results are very confusing and their usefulness in the development of lignocellulosic biorefineries. This manuscript is generally well written and clearly presented however still needs to address some comments, and thus require moderate revision to improve the quality of the manuscript.
1.Title should modify which can describe the whole research work how authors can say it is simple?
Re: We sincerely appreciate the valuable comments. We have removed the word "simple" from the title.
- Abstract looks very general. Need to rewrite in precise way and at the end mention the importance of research work briefly.
Re: Thank you for your suggestion. We have rewritten the abstract according to your request, as shown in the text marked in red.
- In the introduction discuss briefly about recent advances for the detection of dopamine with their pros and cons.
Re: Thank you for your suggestion. We have added recent advances for the detection of dopamine with their pros and cons in the introduction section, as shown in the text marked in red.
- Optimization studies results should be in the main text.
Re: Thank you for your suggestion. We have moved the optimization studies results from the supporting document into the main text, as shown in the text marked in red.
- Is there any effect of temperature and pH on the results?
Re: As can be seen from Figure S3, the UV absorption value increases with the increase of PH. When PH is 7, the UV absorption value reaches the maximum; when PH is 8, the UV absorption value greatly decreases to 0.4. The UV absorption value increases with the increase of temperature, and reaches the maximum when the temperature is 45℃. After increasing the temperature, the UV absorption value does not increase and slowly decreases.
- ls there any morphological changes that can be observed with the developed material?
Re: As shown in Fig. 2(c), The CuS@PB/Pt composite recovered its flower shape, and small particles were present on the nanosheets.
Figure 2. SEM images of (a) CuS (b) CuS@PB (c) CuS@PB/Pt; (e) XPS spectra of the CuS@PB/Pt; XPS spectra of (f) C 1S, (g) Cu 2p, (i) S 2p, (j) Pt 4f, (k)Fe 2p; (d)(h)(l) The corresponding EDS elemental mapping images of the prepared CuS@PB/Pt including the S, Cu, Fe, and Pt, scale bar: 2 nm.
Statistical analysis of the results should be provided in the materials and methods section. lt's important for all experimental work Report these values in the resultsand discusslon
Re: The statistical analysis of the results were usually be provided in the result section, just like the following literature.( https: / / doi.org/10.1016/j.electacta.2022.141535, https://doi.org/10.1016/j.cej.2022.134681, : https://doi.org/10.1039/d2an01420c )
- Compare your results with other studies by adding one table for the same.
Re: Thank you for your suggestion. We have added a table(Table1) for Comparison of different methods for detection of dopamine., as shown in the text marked in red.
Table 1 Comparison of different methods for detection of dopamine.
|
Method |
Materials |
LOD |
Sensitivity |
Ref |
|
HPLC-MS/MS |
/ |
1.87μM |
1.87× 10−5-1.87μM |
[9] |
|
Electrochemical |
Co3O4–Fe2O3 |
0.24 µM. |
10-100 µM |
[29] |
|
Fluorescence |
2, 3-diaminophenazine |
1.76 μM |
2.0 - 61 μM |
[13] |
|
Colorimetric |
MVCM |
0.74μM |
5-100 μM |
[17] |
|
Colorimetric |
CuS@PB/Pt |
0.28μM |
1 - 60 µM |
This work |
- Techno Economic challenges and limitations of this system should be included? Add future research directions also.
Re: Thank you for your suggestion. We have added what you mentioned above to the conclusion,as shown in the text marked in red.
- The conclusion of the study needs to be added with the specific output obtained from the study, it could be modified with precise outcomes with a take home message.
Re: Thank you for your suggestion. We have revised our conclusions with precise outcomes with a take home message, as shown in the text marked in red.
10.Some English and grammar mistakes are present that need to be correct to improve the quality of the manuscript.
Re: We sincerely thank the reviewer for careful reading. In our resubmitted manuscript, the typo is revised. Thank you for your correction.

Round 2
Reviewer 2 Report
Comments and Suggestions for Authors
see attached file

Author Response
Review comments
The authors partially answered my questions.
- The introduction was not expanded as indicated. Including the discussion on precursors as future research does not make much sense because this type of research already exists. My suggestion was to provide a broad overview of the importance of investigating these molecules to support the work you have done.
Re: Thank you for your suggestion. We have expanded on the importance of investigating these molecules in introduction, as shown in the text marked in red.
- There continue to be discrepancies in the figures. In my opinion, it must have similar graphics throughout the manuscript. (E.g. different wording between figures 1, 2, 3, etc)
Re: Thanks for your careful checks. We are sorry for our carelessness. Based on your comments, we have unified the graphics throughout the manuscript.
- Better explain the role of H2O2 in section 3.2. Why the same system without H2O2 has no absorbance and with H2O2
Re: Thank you for your suggestion. We've explained the role of H2O2 in section 3.2 , as shown in the text marked in red.
- What is the concentration used for the analysis in section 3.3?
Re: CuS@Pb/Pt (1 mg/mL, 30 μL) ;PBS buffer (0.01 M, 320 μL) ;TMB (50mM ,110 μL) ; H2O2 (50 mM, 50μL).
- Error bars are missing in fig. 4.
Re: Thanks for your careful checks. There are error bars in Figure 4b. However, due to the small error of results, it can’t be clearly displayed. We have modified the format of Figure 4 that it can be seen more clearly. Thank you very much for your reminding.
- Section 3.4 shows the calibration curves of the dopamine response but the UV-Vis spectrum is missing. In fact, if I were to compare these lines with the curves shown in figure 3c, neither the value at zero dopamine concentration nor the value at the addition of 10 and 100 mM would match.
Re: As shown in Figure 3(c), after adding different amounts of DA, the absorption peak intensity of ox-TMB stabilized at 652nm, so all our tests were conducted at 652nm. According to your reminder, we have re-collated and checked the data of 3.2 Feasibility of the designed colorimetric strategy, and we found that the datas of reaction temperature and temperature were wrong. We have already changed and marked red in the text. We apologize for the mistake.
- Figure 6 is badly done. You can perceive error bars but you do not understand them. Among other things, some do and some do not. Here too, the writing of the figure is different from the others (see for example the letters a,b,c which are gigantic compared to those used in the previous figures).
Re: Thanks for your careful checks. We have we have unified the size of the letters in the figure. The figure6(c)and(g) were the actual measurement for 7 times, so there is no error bar.
- You can see from Figure 6d that AA gives a good response. Have you done dopamine detection tests in the presence of AA? How does it behave?
Re: We supplemented the experiment and it can be seen from Figure 6 (b) and (f) that when DA and AA are present at the same time, their absorption values are affected. Therefore, we introduced a masking agent iodoacetamide (IAM) to address this issue, the introduction of IAM shields the interfering signal response well, thereby improving the selectivity of the sensor for DA.
- Line 194 (1-60 M)???? seems to be wrong.
Re: Thanks for your careful checks. We are sorry for our carelessness. Based on your comments, we have unified the graphics throughout the manuscript.
- How was the LOD calculated?
Re: LOD=3s/s (Where s represents the standard deviation after 11 parallel tests in the blank DA, and s represents the slope of the calibration curve).
- Section 3.5 still different graphics in the figure. Also the error bars are not visible.
Re: Thanks for your careful checks. As you can see, Figure 7(b) and (c) are not curves of the concentration-response relationship, so there are not error bars.
